# Comparison of Mandibular Volume and Linear Measurements in Patients with Mandibular Asymmetry

**DOI:** 10.3390/diagnostics13071331

**Published:** 2023-04-03

**Authors:** Yuki Hikosaka, So Koizumi, Yong-Il Kim, Mohamed Adel, Mohamed Nadim, Yu Hikita, Tetsutaro Yamaguchi

**Affiliations:** 1Department of Orthodontics, School of Dentistry, Kanagawa Dental University, Yokosuka 238-8580, Japan; 2Department of Orthodontics, Pusan National University Dental Hospital, Yangsan 50612, Republic of Korea; 3Division of Orthodontics, College of Dentistry, University of Kentucky, Lexington, KY 40536, USA; 4Department of Orthodontics, Suez Canal University, Ismailia 41522, Egypt

**Keywords:** mandibular asymmetry, cone-beam computed tomography, mandibular body, ramus, condyle

## Abstract

In patients with mandibular asymmetry, the volume of the mandible divided by the mandibular median plane is significantly larger on the non-deviated (N-Dev) side than on the deviated (Dev) side. However, it has been reported that there is no significant difference between the volumes of the N-Dev and Dev sides when the mandibular ramus and body are divided. The purpose of this study was to investigate which region is responsible for the volume difference between the N-Dev and Dev sides. Cone Beam Computed Tomography (CBCT) images of patients with mandibular asymmetry were analyzed by measuring the volume, and linear analysis of the mandibular body, ramus, and condyle on the N-Dev and Dev side was performed. In this study, CBCT images of 37 patients (8 Japanese, 16 Korean, and 13 Egyptian) aged ≥ 18 years with mandibular asymmetry (men: 20, women: 17) were used to evaluate mandibular asymmetry. In patients with mandibular asymmetry, the N-Dev side showed significantly larger values than the Dev side for both volume and linear condyle, ramus, and mandibular body measurements. These results do not differ according to sex or ethnicity. Therefore, it is suggested that the N-Dev side of mandibular asymmetry is large without any regional specificity in pathophysiology.

## 1. Introduction

Maxillofacial asymmetry is an important issue in orthodontic treatment and one of the chief complaints of patients seeking orthognathic surgery. Facial asymmetry is often influenced by mandibular deviations [1]. The etiology of facial asymmetry includes congenital disorders, acquired diseases, and traumatic and developmental deformities. Since remodeling by endochondral ossification of the condyle and addition of bone are the major growth contributors to the mandible, infection or failure of the condylar cartilage is also believed to induce asymmetry [2]. It is also known that muscle activity affects facial asymmetry, with cases with unilateral posterior crossbite differing between crossbite and non-crossbite sides. While subjects with unilateral posterior crossbite show smaller occlusal forces than non-crossbite subjects, there is no consistency in studies reporting asymmetry in the thickness of the masticatory muscles of the subjects [3].

The prevalence of mandibular asymmetry by ethnicity and gender has also not been fully investigated [4].

It has been shown that ethnicity and gender differences have no effect on the perception of asymmetry [5,6]. Although studies have demonstrated racial differences in dental anatomy [7] and cranial shape [8], the role of ethnicity in the pathogenesis of mandibular asymmetry is not well understood. It has been reported that factors influencing the subjective evaluation of the degree of facial asymmetry include the nasal area, angle of mouth tilt, chin deviation, and left–right difference in gonial angle [9]. However, in orthodontic diagnosis, it is essential to differentiate whether the facial asymmetry is dental, mainly due to alveolar and dental problems, or skeletal, mainly due to abnormal mandibular morphology and position, and radiographic evaluation is required. Traditionally, the diagnosis of facial asymmetry has been based on two-dimensional assessments, such as distance and angle assessments in frontal cephalograms [10]. By contrast, Cone Beam Computed Tomography (CBCT) has become more common in recent years to provide a more detailed three-dimensional view of the pathophysiology of mandibular asymmetry in addition to a planar two-dimensional evaluation [11]. In patients with mandibular asymmetry, the volume of the mandible divided by the mandibular midplane is significantly larger on the non-deviated (N-Dev) side than on the deviated (Dev) side [12]. However, when the mandible was divided into ramus and mandibular body and the volume of the N-Dev and Dev side were examined, there was no significant difference between them [13]. In this study, we evaluated the linear and volumetric characteristics of the mandible using CBCT images of patients diagnosed with mandibular asymmetry undergoing orthodontic treatment to determine the region responsible for the deviation, frequency of deviation, and differences according to ethnicity and sex.

## 2. Materials and Methods

### 2.1. Participants

From a total of 319 patients (72 Japanese, 88 Koreans, 159 Egyptians; 132 men, 187 women), 37 patients (8 Japanese, 16 Koreans, 13 Egyptians; 20 men, 17 women) aged ≥ 18 years with mandibular asymmetry who met the selection criteria were selected for the study. The Japanese Participants were eight orthodontic patients (men: 3; women: 5) from Kanagawa Dental University Hospital and were approved by the Kanagawa Dental University Ethics Committee (approval numbers 642 and 663). The Korean Participants were 16 (men:11; women:5) orthodontic patients from Pusan National University Dental Hospital and the study was approved by the Institutional Review Board of Pusan National University Dental Hospital (IRB PNUDH-2019-025). The Egyptian Participants were 13 (men: 6; women: 7) orthodontic patients from the Suez Canal University and were approved by the Ethics Committee of the Suez Canal University (IRB 8). The sample size was calculated using a *t*-test by G*Power software (ver. 3.1.9.6; Heinrich-Heine-Universität Düsseldorf, Düsseldorf, Germany) according to measurements of the non-deviated and deviated side groups of the mandibular body length (90.6 ± 5.3 mm and 93.3 ± 4.5 mm, respectively) from a previous study [13]. The effect size was set at 0.5, the α error was 0.05, and the power was 0.8. The sample size needed for this study was set at 35 participants. All CBCT data used in this study were obtained for orthodontic treatment and not for research purposes. This study was conducted in accordance with the Declaration of Helsinki, and informed consent was obtained from all patients prior to the start of the study. The report is per the current standards recommended for reporting observational studies in epidemiology (STROBE).

Selection criteria for study participants: (1)Patients who were at least 18 years old.(2)Patients with a Menton (Me) deviation ≥ 4.0 mm or more from the midsagittal plane (MSP) of the face.

Exclusion criteria for this study:(1)Patients with pre-existing congenital diseases, such as cleft lip or palate.(2)History of orthodontic treatment or orthognathic surgery.(3)History of maxillofacial trauma.(4)CBCT images of inadequate quality owing to artifacts.(5)CBCT images with inadequate coverage.

### 2.2. Data Acquisition and 3D Landmarks

CBCT images of the Japanese patients were acquired using a CBCT system (KaVo 3D eXam, KaVo, Biberach, Germany), CBCT images of the Korean patients were acquired using a cone-beam X-ray CT system (Zenith3D, Vatech Co., Seoul, Republic of Korea), and those of Egyptian patients were obtained using a cone-beam X-ray CT system (Soredex SCANORA 3D, Nahkeatine 16; TUUSULA, Finland). All CBCT data were stored in Digital Imaging and Communications in Medicine (DICOM) and reconstructed as 3D images after input into the imaging software Invivo™ 6 (Anatomage, San Jose, CA, USA). Thirteen 3D landmarks were selected for measurement according to procedures used in previous studies [14] (Table 1). The Frankfurt Horizontal Plane (FHP) was defined as the plane passing through the bilateral polion (Po) and left orbit (Or), and the facial Midsagittal Plane (MSP) as the plane passing perpendicular to the FHP through the nasion (N) and sera (S) (Figure 1). The side with a lateral menton (Me) deviation relative to the MSP was defined as the Dev side, and the opposite side was defined as the N-Dev side. The plane connecting Me, point B, and the mental spine (G) was defined as the Absolute Mandibular midsagittal Plane (AMP), and the mandible was divided into N-Dev and Dev sides by the AMP (Figure 2). The volume of the mandibular body, ramus, and condyle, and five linear analysis parameters were measured on both the N-Dev and Dev sides.

### 2.3. Measurements

Measurements were performed by a single researcher. To evaluate the error of the measurer, 30 CBCT images were randomly selected and measured at two-week intervals under the same conditions using Dahlberg’s equation [15].

Linear measurements were performed according to the method described by Hasebe et al. [16] and Al-koshab et al. [17]. Ramus length was defined as the distance between the uppermost point of the condyle (Cd) and the gonion (Go), and mandibular body length was defined as the distance between Go and Me (Figure 3). The length of the condyle was measured as the distance from the plane passing through the lowest point of the sigmoid notch (InfSig) perpendicular to the ramus plane in the sagittal plane and from Cd (Figure 4).

The volumes of the mandibular body and ramus were measured according to the method by Kwon et al. [13]. Condylar volume was measured as described by Al-Koshab et al. [17], Chang, et al. [18], and Huang, et al. [19]. For the volume of the mandible, the Dev side and the N-Dev side were compared, and the mandible was divided into three segments (Figure 5, Figure 6 and Figure 7): mandible body, ramus, and condyle, as per a previous study [18,19]. The mandible was divided into the left and right sides by the AMP. Furthermore, the mandible body and ramus were divided by the plane connecting Go, junction lateral (Jlat), and junction medial (Jmed). The difference between the measured volume of the N-Dev and the Dev side was calculated for each using the N-Dev side minus the Dev side volume. Measurements of the mandibular body, ramus, and condyle volumes were obtained using the Invivo™ 6 imaging software using the following procedure. First, a 3D model of the mandible was created from DICOM images with the threshold set to the software’s preset value of ‘‘bone”. The trimming function was then used to cut out the three types of measurement portions, and the volume was measured using an automatic measurement function. As CBCT image data obtained from three different CBCT systems were used in this study, prior to the measurements, we verified using aluminum bars that the differences in CBCT systems did not affect the measured values, referring to a previous study by Katayama et al. [20].

### 2.4. Statistical Analysis

Statistical analyses were performed using the statistical analysis software SPSS Statistics 26 (IBM Corporation, Armonk, NY, USA). Corresponding *t*-tests were used for statistical analysis. Volumes were evaluated using a one-sample *t*-test to determine the difference between Dev and N-Dev side values. Repeated-measures analysis of variance (ANOVA) was used to adjust for sex and ethnicity differences and to check for the presence of interaction statistics. Statistical significance was defined as *p* < 0.05.

## 3. Results

The results of linear measurements are listed in Table 2. The Dev side was significantly longer than the N-Dev side in all three measurement areas: condyle length, ramus length, and mandibular body length (mandibular body length: Dev side—84.6 ± 7.2; N-Dev side—87.4 ± 7.3; ramus length: Dev side—59.6 ± 6.7, N-Dev side—63.4 ± 7.1; condyle length: Dev side—22.5 ± 4.4, N-Dev side—25.6 ± 4.9, *p* < 0.001). The comparison test adjusted for differences in sex and population (Table 3 and Table 4) showed a significant difference between the Dev and N-Dev sides (*p* < 0.001). However, since there was no interaction between each factor and sex or population (*p* > 0.05), it was concluded that the trend of the difference did not differ by sex or population.

The results of volumetric measurements are presented in Table 5. The Dev side was significantly larger than the N-Dev side in all four measurement sites: hemi-mandibular, condyle, ramus, and mandibular volume (hemi-mandibular volume: [Dev side—N-Dev side] 2.7 ± 4.1, *p* < 0.001; mandibular body volume: [Dev side—N-Dev side] 1.5 ± 4.2, *p* = 0.030; ramus volume: [Dev side—N-Dev side] 1.2 ± 2.3, *p* = 0.004; condyle volume: [Dev side—N-Dev side] 0.6 ± 1.1, *p* = 0.002). When the sex differences were adjusted for (Table 6), there was a significant difference in volume between the Dev and N-Dev sides (*p* < 0.01). However, since there was no interaction between each factor and sex (*p* > 0.05), it was concluded that the trend of the difference did not differ by sex (Table 6). When the comparison test was adjusted for differences in the population (Table 7), significant differences were found between the Dev and N-Dev sides for the condyle and ramus volumes *p* < 0.01). When the comparison test was performed adjusting for differences in population, the mandibular body volume showed no statistically significant difference between the Dev and N-Dev side volumes (*p* = 0.059), but the N-Dev side volume showed a trend toward greater volume. However, since there was no interaction between the two populations (*p* > 0.05), it was concluded that the trend of the difference between the two populations was not different (Table 7).

## 4. Discussion

In this study, we extracted and analyzed mandibular deviation cases from CBCT data of 319 Japanese, Korean, and Egyptian patients. Thirty-seven cases of facial asymmetry were extracted from 319 patients. The proportion was 11.6%, which is similar to that reported in a previous study [21], although it depends on the definition of asymmetry.

Since the reference plane used in this study (AMP) is defined on a landmark basis, it is easy to establish and less prone to measurement error for the quantitative comparison of volumes and distances, and may be useful for pathological evaluation when performing orthognathic treatment in patients with jaw deformity and deviation. Furthermore, the results of this study may help in clarifying the risk factors for bad fracture, which is one of the complications of osteotomy, and in considering the method of fixation when the volume of the mandible and other factors differ in orthognathic surgery.

In this study, linear and quantitative evaluations of the mandible were performed to determine which region was responsible for the deviation, and the N-Dev side was found to be significantly larger than the Dev side in both volume and linear analysis measures in all regions, without region specificity in the pathology of mandibular asymmetry. Some previous studies have shown that the volume of the mandible divided by the mandibular midplane is significantly greater on the N-Dev side than on the Dev side [12], whereas in other studies, in patients with mandibular asymmetry, it has been reported that there is no significant difference in N-Dev and Dev side volumes when the mandibular prominence and body are divided [13]. These differences in results may be due to the limitation that many studies, including the present study, evaluated only the mandible with respect to the reference plane, and the maxillary region was excluded from the evaluation.

No effect of gender or population interaction was observed. On the other hand, a repeated-measures ANOVA conducted to adjust for population differences showed no difference between the Dev and N-Dev sides in mandibular body volume. Sample size may have influenced this result. It is also possible that the presence of a Stafne defect, a bone defect that reduces mandibular volume [22], may have affected the results.

The presence or absence of maxillary deviation affects the mandible, which lies below the maxilla, when assessing deviation [23,24]. Further studies are needed to determine the extent of maxillary deviation.

Interestingly, the present study showed that leftward mandibular deviation was more common regardless of sex and ethnicity (Table 1). It has been known for some time that left-sided deviation is more common than right-sided deviation [25,26,27]. Although the relationship with the dominant hand has been considered a possible reason for the prevalence of left-sided deviation [26], the underlying causes are not yet clear. It is also known that the left side of the human face is more expressive of emotions than the right side [28]. This is because the right side of the brain is more in charge of emotional control than the left side, and facial expressions are more richly expressed on the left side, where the right side of the brain is dominant [28]. Therefore, people often unconsciously turn the left side of their faces toward others, and it is said that the left side of the face is often depicted in portraits. On the other hand, it is also known that not only the person making the expression but also the observer has a left–right difference in the evaluation of deviation, and that the left side is evaluated to have milder deviation than the right side, even if the left side deviates by the same amount as the right side [29,30]. It is also known that there is a difference in masticatory muscle activity between the Dev and non-mutated sides, with the Dev side showing higher muscle activity [31]. It is known that muscle activity, not only of the masticatory muscles but also of the facial muscles, is higher on the Dev side [31,32,33,34]. Thus, there are various possible reasons for the deviation and the left–right difference, and further research is required.

In addition, mandibular growth is enhanced by the posterior and upward growth and development of the ramus. The medial pterygoid and masseter muscles are attached to the medial and lateral sides of the gonial angle of the mandible, respectively, and the more active the muscles are, the more the gonial angle develops. Remodeling, including mandibular growth, continues throughout life in relation to soft-tissue function. Moss suggested this phenomenon as a functional matrix hypothesis [35].

Previous studies [31] have shown that the activity of one side of the masticatory muscles affects the difference in growth between the left and right sides of the mandible. In the 3D evaluation using computed tomography for cases of deviation, the Dev side had a shorter bone body, ramus, and a smaller gonial angle [36], and the same results were observed in this study.

There are several types of mandibular braiding: congenital or hereditary, such as hemifacial microsomia, functional lateral mandibular deviation, as seen in maxillary dental stenosis, growth suppression, or condylar hyperplasia due to condylar injury [2,37,38,39,40,41]. In this study, we excluded the cases of deviation due to congenital diseases. The fact that not only ramus height but also mandibular length and volume were smaller on the Dev side regardless of ethnicity or sex suggests that growth inhibition due to condyle trauma alone cannot explain the cause of the deviation, and the left–right difference in growth and development via the difference in left–right muscle activity caused by some reason is a major factor in causing the deviation.

This study had some limitations, and they include the fact that muscle activity was not measured and that it was a cross-sectional study. The muscle activity-affected deviation requires assessment of muscle activity and evaluation with longitudinal studies over time. Furthermore, although this study was conducted in patients who were 18 years and older, quantification of mandibular asymmetry at various time points in growing patients would provide clinically relevant research results.

## 5. Conclusions

The results of this study suggest that the N-Dev side of mandibular asymmetry is large without region specificity. Furthermore, there are no sex differences or ethnicity differences in mandibular asymmetry. These findings could contribute to understanding the risk factors for bad fracture, which is one of the complications of osteotomy, and to considering the method of fixation when the volume of the mandible and other factors differ in orthognathic surgery.

## Figures and Tables

**Figure 1 diagnostics-13-01331-f001:**
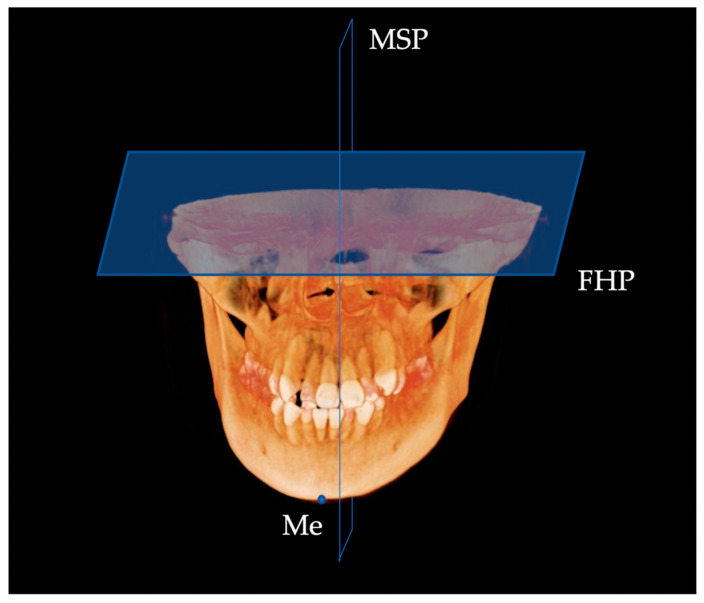
Facial midsagittal plane. MSP, midsaggital plane; Me, menton; FHP, Franfurt Horizontal Plane.

**Figure 2 diagnostics-13-01331-f002:**
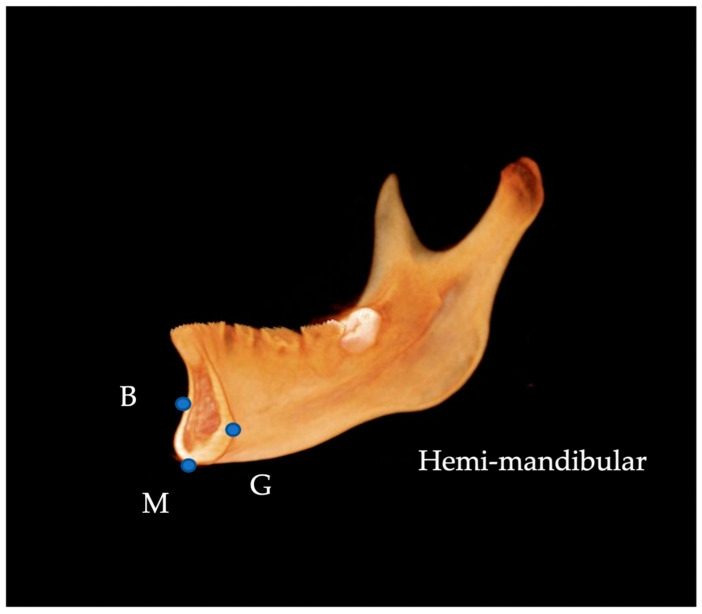
Segmentation of the mandible. M, menton; G, genial tubercle; B, supramentale.

**Figure 3 diagnostics-13-01331-f003:**
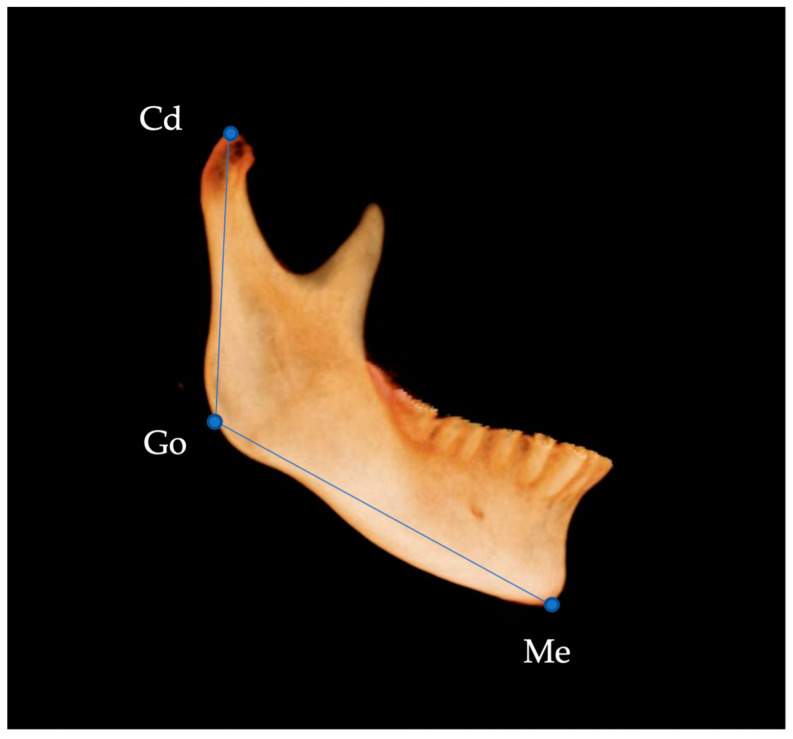
Length of mandibular body and ramus. Cd, uppermost point of the condyle; Go, gonion; Me, menton.

**Figure 4 diagnostics-13-01331-f004:**
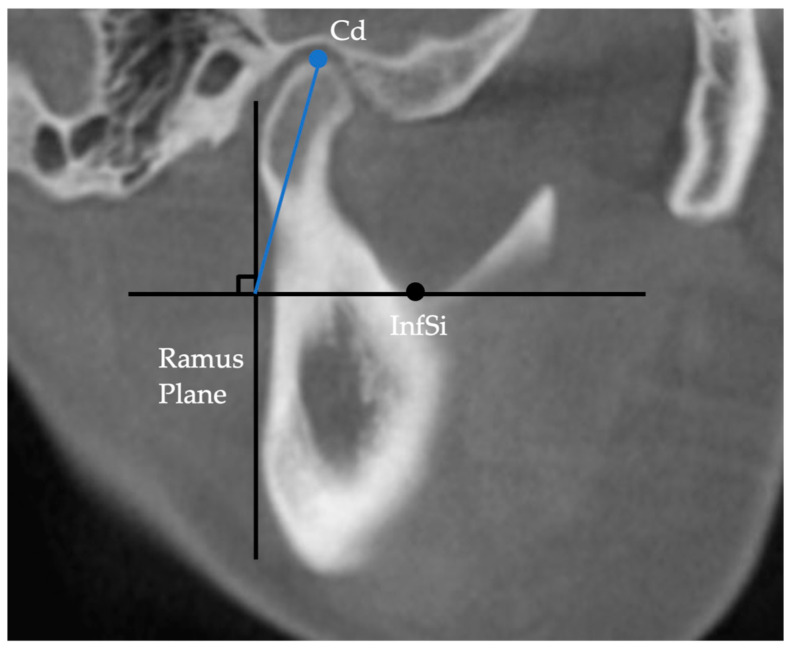
Facial midsagittal plane. InfSig, lowest point of the sigmoid notch; Cd, uppermost point of the condyle.

**Figure 5 diagnostics-13-01331-f005:**
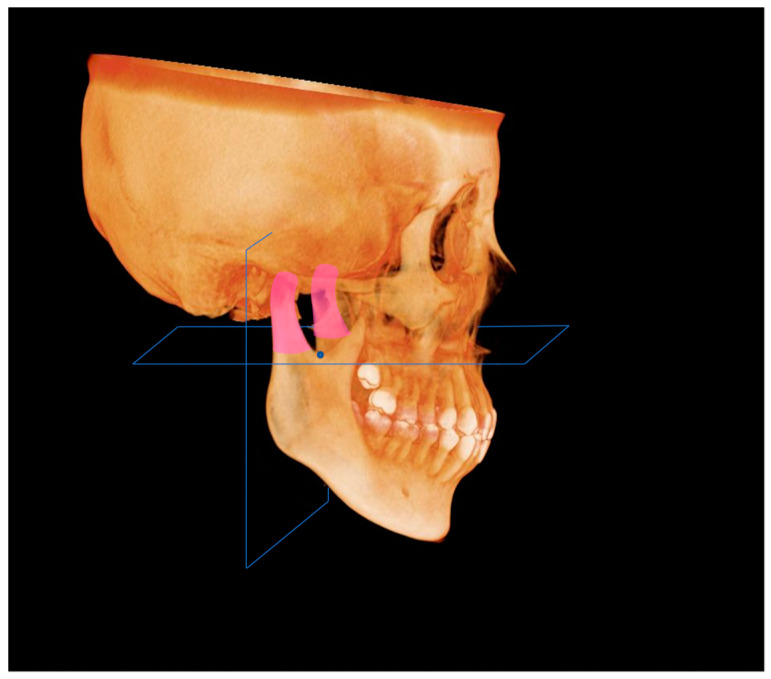
Volume of the condyle.

**Figure 6 diagnostics-13-01331-f006:**
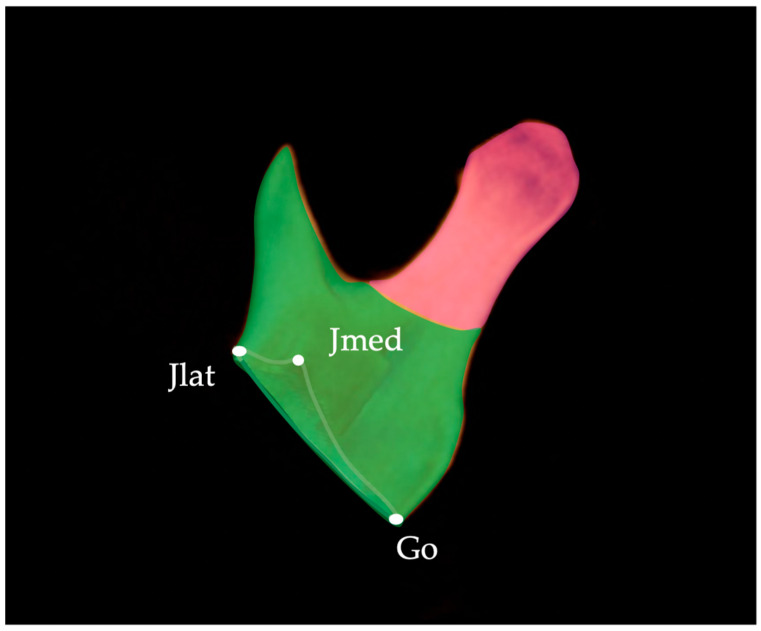
Volume of the ramus. Jlat, junction lateral; Jmed, junction medial; Go, gonion.

**Figure 7 diagnostics-13-01331-f007:**
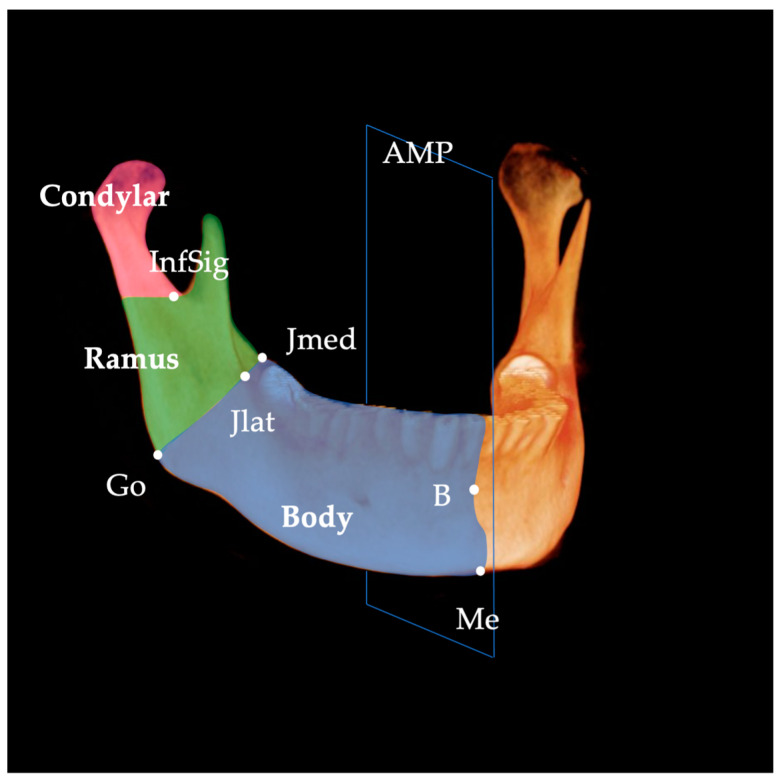
Three divisions of the mandible. Infsig, lowest point of the sigmoid notch; Me, menton; B, supramentale; Jlat, junction lateral; Jmed, junction medial; Go, gonion.

**Table 1 diagnostics-13-01331-t001:** Three-dimensional landmarks.

Landmark		Definition
N	nasion	Middle point of nasofrontal suture
S	sella	Centre of sella turcica
Or	orbitale	Most inferior point of the lower margin of the orbit
Po	porion	Most superior point of the external auditory meatus
Con	condylion	Most superior point of the condylar head
Jlat	junction lateral	Most lateral and deepest point of curvature formed at the junction of the mandibular ramus and body
Jmed	junction medial	Most medial and deepest point of curvature formed at the junction of the mandibular ramus and body
Go	gonion	Most inferior, posterior, and lateral point on the external angle of the mandible
Me	menton	Most inferior midpoint on the symphysis
Pog	pogonion	Most anterior midpoint on the symphysis
B	supramentale	Midpoint of greatest concavity on the anterior border of the symphysis
G	genial tubercle	Midpoint on genial tubercle
InfSig	inferior sigmoid	Most inferior point of the sigmoid notch

**Table 2 diagnostics-13-01331-t002:** Linear measurements.

	Mandibular Body Length (mm)			Ramus Length (mm)			Condyle Length (mm)		
Dev side	84.6 ± 7.2	M: 87.7 ± 6.4	J: 81.1 ± 7.0	59.6 ± 6.7	M: 63.1 ± 5.6	J: 58.7 ± 5.5	22.5 ± 4.4	M: 23.0 ± 4.7	J: 25.0 ± 3.8
		F: 80.9 ± 6.3	K: 89.4 ± 6.2		F: 55.5 ± 5.6	K: 63.1 ± 5.5		F: 21.8 ± 4.2	K: 22.7 ± 4.2
			E: 80.8 ± 4.9			E: 55.8 ± 6.9			E: 20.6 ± 4.4
N-Dev side	87.4 ± 7.3	M: 91.0 ± 6.2	J: 84.4 ± 7.7	63.4 ± 7.1	M: 67.1 ± 5.5	J: 64.2 ± 5.2	25.6 ± 4.9	M: 26.2 ± 4.7	J: 28.8 ± 3.9
		F: 83.0 ± 6.2	K: 92.7 ± 5.6		F: 59.1 ± 6.5	K: 67.1 ± 5.7		F: 24.8 ± 5.0	K: 26.3 ± 4.8
			E: 82.6 ± 4.2			E: 58.4 ± 7.1			E: 22.7 ± 4.1
*p*	<0.001			<0.001			<0.001		

M, men; F, women; J, Japanese; K, Korean; E, Egyptian.

**Table 3 diagnostics-13-01331-t003:** Repeated-measures analysis of variance for adjustment of sex for line measurements.

Factor	SS	df	MS	F	*p*
Condyle	174.395	1	174.395	28.434	0.000
Condyle × sex	0.102	1	0.102	0.017	0.898
Error	214.669	35	6.133		
Ramus	264.337	1	264.337	28.894	0.000
Ramus × sex	0.681	1	0.681	0.074	0.787
Error	320.193	35	9.148		
Mandibular body	134.577	1	134.577	55.113	0.000
Mandibular body × sex	5.742	1	5.742	2.352	0.134
Error	85.464	35	2.442		

SS, sum of square; df, degree of freedom; MS, mean square.

**Table 4 diagnostics-13-01331-t004:** Repeated-measures analysis of variance for adjustment of population for line measurements.

Factor	SS	df	MS	F	*p*
Condyle	167.697	1	167.697	27.849	0.000
Condyle × Population	10.040	2	5.020	0.834	0.443
Error	204.732	34	6.022		
Ramus	271.593	1	271.593	30.695	0.000
Ramus × Population	20.042	2	10.021	1.133	0.334
Error	300.832	34	8.848		
Mandibular body	131.338	1	131.338	55.506	0.000
Mandibular body × Population	10.756	2	5.378	2.273	0.118
Error	80.451	34	2.366		

SS, sum of squares; df, degree of freedom; MS, mean square.

**Table 5 diagnostics-13-01331-t005:** Volumetric measurements.

Volumetric Measurements (mm^3^)	Hemi-Mandibular		Mandibular Body		Ramus		Condyle	
N-Dev side—Dev side	2.7 ± 4.1 (*p* < 0.001)	M: 2.3 ± 3.9	1.5 ± 4.2 (*p* = 0.030)	M: 0.8 ± 3.6	1.2 ± 2.3 (*p* = 0.004)	M: 1.4 ± 1.3	0.6 ± 1.1 (*p* = 0.002)	M: 0.6 ± 0.7
		F: 3.3 ± 4.4		F: 2.3 ± 4.7		F: 1.0 ± 3.2		F: 0.6 ± 1.5
		J: 1.6 ± 4.2		J: 0.1 ± 3.9		J: 1.5 ± 1.4		J: 0.5 ± 0.7
		K: 3.2 ± 3.5		K: 1.2 ± 2.2		K: 2.0 ± 2.2		K: 0.9 ± 1.5
		E: 2.9 ± 4.1		E: 2.9 ± 5.8		E: 0.0 ± 2.6		E: 0.3 ± 0.7

M, men; F, women; J, Japanese; K, Korean; E, Egyptian.

**Table 6 diagnostics-13-01331-t006:** Repeated-measures analysis of variance for adjustment of sex for volume measurements.

Factor	SS	df	MS	F	*p*
Condyle	6.954	1	6.954	10.821	0.002
Condyle × sex	0.007	1	0.007	0.012	0.915
Error	22.494	35	0.643		
Ramus	25.545	1	25.545	9.087	0.005
Ramus × sex	0.939	1	0.939	0.334	0.567
Error	98.396	35	2.811		
Mandibular body	47.526	1	47.526	5.470	0.025
Mandibular body × sex	9.877	1	9.877	1.137	0.294
Error	304.089	35	8.688		
Hemi-mandibular	141.900	1	141.900	16.549	0.000
Hemi-mandibular × sex	4.662	1	4.662	0.544	0.466
Error	300.101	35	8.574		

SS, sum of squares; df, degree of freedom; MS, mean square.

**Table 7 diagnostics-13-01331-t007:** Repeated-measures analysis of variance for adjustment of population for volume measurements.

Factor	SS	df	MS	F	*p*
Condyle	5.674	1	5.674	9.044	0.005
Condyle × Population	1.172	2	0.586	0.934	0.403
Error	21.330	34	0.627		
Ramus	23.400	1	23.400	9.553	0.004
Ramus × Population	16.055	2	8.028	3.277	0.050
Error	83.279	34	2.449		
Mandibular body	32.907	1	32.907	3.830	0.059
Mandibular body × Population	21.836	2	10.918	1.271	0.294
Error	292.130	34	8.592		
Hemi-mandibular	110.995	1	110.995	12.682	0.001
Hemi-mandibular × Population	7.195	2	3.598	0.411	0.666
Error	297.569	34	8.752		

SS, sum of squares; df, degree of freedom; MS, mean square.

## Data Availability

The data presented in this study are available from the corresponding author upon request.

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
