# Peer review of "Comparison of Mandibular Volume and Linear Measurements in Patients with Mandibular Asymmetry"

_diagnostics, 2023, doi:10.3390/diagnostics13071331_

Round 1

Reviewer 1 Report

1. The sample size is too small and there is no strong control group.

2. The specific differences between races are not shown.

3. Authors should clarify the clinical significance.

Author Response

We thank the Reviewer for providing a thorough review of our manuscript, and for making valuable suggestions for change. We have responded to each item in the Reviewer’s critique and have modified the manuscript accordingly.

Reviewer 1:

  1. The sample size is too small and there is no strong control group.

Thank you for your suggestion. The sample size for this study was determined by setting the effect size to 0.5 from (90.6 ± 5.3, 93.3 ± 4.5) based on the following reference [1].

The post-hoc test for linear analysis also yielded a power of 0.67~0.97. On the other hand, the power for the mandibular body was 0.67. Furthermore, the repeated measures analysis of variance (Table 7), which confirmed the population interaction, showed no significant difference. As you indicated, this may have been influenced by the insufficient sample size. Therefore, we have noted the possibility that the lack of samples may have affected the results of the discussion.

This study was designed with reference to the following literature [1,2], and we evaluated the linear and volumetric characteristics of the mandible with mandibular asymmetry to explore the region. Therefore, we did not establish a control group without mandibular deviation. Thank you for your understanding.

1) Similarity index for intuitive assessment of three-dimensional facial asymmetry. Sci Rep. 2019 Jul 29;9(1):10959.

2) Diagnosis and Surgical Outcomes of Facial Asymmetry According to the Occlusal Cant and Menton Deviation. J Oral Maxillofac Surg. 2019 Jun;77(6):1261-1275.

  1. The specific differences between races are not shown.

 Thank you for your comments.

 In this study, we used repeated measures analysis of variance to examine the interaction effect of race on the difference between the deviated and unevolved sides. Only the analysis of mandibular body volume was not significantly different (Table 7). This result may have been influenced by sample size and has been added to the discussion.

3. Authors should clarify the clinical significance.

 Thank you for your suggestion.

 We have added a comment on clinical significance to the discussion.

Reviewer 2 Report

This study investigated the mandibular volume in people with mandibular asymmetry. Overall, the study was well conducted. There are no major concerns, only a few comments to improve the paper. 

Did the authors observe any effect of age on the parameters studied? Age-related morphometric changes of bone parameters are not uncommon. This is the reason why there are numerous studies on bone morphometry and age prediction. 

The authors should be aware of Stafne defect, a bone defect that might reduce the volume of the mandible. This may be added to the discussion. Unlike the defects mentioned between lines 267-169, this defect is asymptomatic. https://www.sciencedirect.com/science/article/pii/S1991790222002057. 

Delete extra “Dev” in line 18. 

Line 220: It would be interesting to mention the inconsistent definition of mandibular asymmetry. How is the definition in this study different from the previous studies? 

Author Response

We thank the Reviewer for providing a thorough review of our manuscript, and for making valuable suggestions for change. We have responded to each item in the Reviewer’s critique and have modified the manuscript accordingly.

Reviewer 2:

This study investigated the mandibular volume in people with mandibular asymmetry. Overall, the study was well conducted. There are no major concerns, only a few comments to improve the paper. 

We thank the Reviewer for the encouraging comments and for the thorough review of our manuscript.

Did the authors observe any effect of age on the parameters studied? Age-related morphometric changes of bone parameters are not uncommon. This is the reason why there are numerous studies on bone morphometry and age prediction. 

Thank you for pointing this out.

We did not evaluate age in this study, as we evaluated bones from adults over 18 years of age. On the other hand, we have added to the discussion that the effect of age is a very significant topic and a subject for future research (P12 L335).

The authors should be aware of Stafne defect, a bone defect that might reduce the volume of the mandible. This may be added to the discussion. Unlike the defects mentioned between lines 267-169, this defect is asymptomatic. https://www.sciencedirect.com/science/article/pii/S1991790222002057. 

Thank you for pointing this out.

I added the possibility that the Stafne defect affected the volume of the mandible.

Delete extra “Dev” in line 18. 

Thank you for pointing this out.

I added "and" to express that "there is no difference between the N-Dev side and the Dev side".

Line 220: It would be interesting to mention the inconsistent definition of mandibular asymmetry. How is the definition in this study different from the previous studies? 

Thank you for your comments.

It has been reported that in patients with mandibular asymmetry, there is no significant difference between the volume of the nondeviated (N-Dev) and deviated (Dev) sides of the mandible when the mandibular ramus and body are divided.

On the other hand, in the present study, all the measured items showed larger values for the N-Dev side than for the deviated side, with no region specificity.
